# Study on Temporal Variations of Surface Temperature and Rainfall at Conakry Airport, Guinea: 1960–2016

**René Tato Loua** [1,2,3,*] ，**Hassan Bencherif** [1,4], **Nkanyiso Mbatha** [5], **Nelson Bègue** [1], **Alain Hauchecorne** [6], **Zoumana Bamba** [2] **and Venkataraman Sivakumar** [4]

1   Laboratoire de l'Atmosphère et des Cyclones, UMR 8105, CNRS, Université de La Réunion, Météo-France, 97490 Réunion, France
2   Centre de Recherche Scientifique de Conakry Rogbane, Conakry 1615, Guinée
3   Direction Nationale de la Météorologie de Guinée, Conakry 566, Guinée
4   School of Chemistry and Physics, University of KwaZulu Natal, Durban 4000, South Africa
5   Department of Geography, University of Zululand, KwaDlangezwa 3886, South Africa
6   Laboratoire Atmosphère, Milieux, Observations Spatiales/Institut Pierre-Simon-Laplace, UVSQ Université Paris-Saclay, Sorbonne Université, CNRS, 78280 Guyancourt, France
*   Correspondence: rene-tato.loua@univ-reunion.fr or lrenetatometeo@gmail.com

**Abstract:** The monthly averaged data time series of temperatures and rainfall without interruption of Conakry Airport (9.34° N 13.37° W, Guinea) from 1960 to 2016 were used. Inter-annual and annual changes in temperature and rainfall were investigated. Then, different models: Mann-Kendall Test, Multi-Linear-Regression analysis, Theil-Sen's slope estimates and wavelet analysis where used for trend analysis and the dependency with these climate forcings. Results showed an increase in temperature with semi-annual and annual cycles. A sharp and abrupt rise in the temperature in 1998 was found. The results of study have shown increasing trends for temperature (about 0.21°/year). A decrease in rainfall (about −8.14 mm/year) is found since the end of 1960s and annual cycle with a maximum value of about 1118.3 mm recorded in August in average. The coherence between the two parameters and climate indices: El Niño 3.4, Atlantic Meridional Mode, Tropical Northern Atlantic and Atlantic Niño, were investigated. Thus, there is a clear need for increased and integrated research efforts in climate parameters variations to improve knowledge in climate change.

**Keywords:** temperature; rainfall; climate indices; wavelet; trend analysis; climate change and Conakry

## 1. Introduction

Climate is naturally variable as evidenced by the irregularity of the seasons from one year to another. Long-term climate variability is of great importance for the estimation of its impact on human activities and for predicting the future climate [1]. The need to develop science programs that, in addition to exploring long-term climate change, can meet the more immediate needs of people and organizations to begin factoring climate risks into planning and management processes [2]. Over the twentieth century, west African region continues to receive a lot of unusual disasters at unexpected moments and areas. This might be a consequence of climate change and then that change is generated on the one hand by anthropogenic activities and on the other hand by natural variation. That's why some previous studies done on West Africa regions highlight the variability of temperature and rainfall and their relationship with climate indices, as Zerbo et al. [3] who studied the relationship between the solar cycle and meteorological fluctuations in West Africa and found that temperature and rainfall are influenced by solar activity. Schulte et al. [4] analysed the influence of climate modes on streamflow in the Mid-Atlantic region of the United States. Many studies also have been done on the intra-seasonal

and inter-annual variability of temperature and rainfall [5–9] over West Africa areas. Previous studies have shown the crucial role of sea surface temperature (SST) anomalies in the tropical Atlantic region. For instance, SST induces forcing on the summer monsoon rainfall over sub-Saharan West Africa [10]. Vizy and Cook [8] highlighted that warm sea surface temperature anomalies influences positively the increase in rainfall along the Guinean coast. In their study on variability of summer rainfall over tropical north Africa during the 1906–1992 period, Rowell et al. [11] showed that the global SST variation are responsible for most of the variability of seasonal (July-August-September) rainfall from 1949 to 1990. Indeed, the annual cycle of rainfall over West Africa depends greatly on SSTs in the Gulf of Guinea [11].

However, any climatological study over West Africa could take into account at least West African Monsoon (WAM) and Inter-Tropical Convergence Zone (ITCZ). It is for the reason aforementioned that several studies have been done on the WAM influence on annual climatic variability in West Africa, [6,12] and its dynamic and onset [13–15]. Furthermore, it was also reported by Nicholson [16,17], that a major role of the WAM system is to transport moisture into West Africa from the Atlantic. In response to the onset of the African monsoon, the upwelling cooling is strongest in the east both because of the strong acceleration of the southerly winds and because the thermocline is shallow there [11].

The inter-annual variability of the WAM is mainly explained by the surface of ocean. It is worthy to note that the surface temperature of the inter-tropical Atlantic can be analysed efficiently. It constitutes an important climatic parameter, in the event of a strong anomaly, in all the coastal areas subjected to the direct impact of the WAM [18]. It is for this reason that Joly and Voldoire. Ref. [12] reported that SST anomalies are maximum in June–July, and are associated with a convective anomaly in the marine ITCZ with a spread over the Guinean coast. ITCZ is the major synoptic-scale system controlling seasonal rainfall [19]. It is well known that the distribution of temperature and rainfall through Earth surface is not homogeneous. Espinoza Villar et al. [20] pointed out the impact of mountain ranges on rainfall and specified that the long-term variability with a decreasing rainfall since the 1980s prevails in June-July-August and September-October-November in the Amazon Basin countries.

Our study area is localized in West Africa, enclosing the three major West African climate zones: Guinean zone (approximately 6–8° N); Soudanian zone (approximately 8–12° N) and Sahelian zone (approximately 12–16° N) [21]. It may be stated that the region of Conakry is part of the Soudanian zone (see Figure 1a). The station of Conakry is located at the international airport of Conakry at 9.34° N and 13.37° W, at 26 m height above the sea level (sl). Given that Conakry is a coastal zone that lies between the Atlantic Ocean and the Kakoulima Mountain range, which forms a barrier and promotes the Foehn phenomenon (see Figure 1b). This feature seems to be the reason that makes it the rainiest area compared to other parts of the country. This coastal site is the national socioeconomic development centre of Guinea, but is always threatened by heavy precipitations and strong heat waves causing significant economic and sanitary damages and loss of lives.

The absolute poverty of a large proportion of the African continent's people renders them highly vulnerable to changes in climate [22]. According to the increasing impact of the climate change in this area and the geo-climatic and environmental factors influences mentioned above, the purpose of our approach is to investigate with the keenest interest the climate variability as well as the forcing led by some climate indices on the temperature and rainfall at Conakry during 57 years. The aim of our study is to improve the understanding and strengthen the knowledge on the climate variability in this region of Guinea through a climatological approach coupled with a digital tool of analysis. After the station dataset description and methodology, obtained results are presented and discussed.

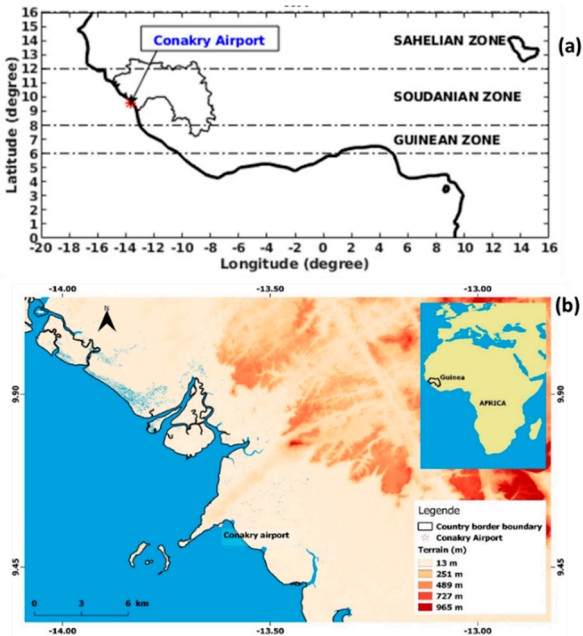

**Figure 1.** Geo-localisation of Conakry station in West Africa, map showing the three major west african climate zones (**a**), map showing Conakry Airport between Atlantic Ocean and Kakoulima Mountain range (**b**).

## 2. Materials and Methods

### 2.1. Data

Monthly averages of temperature and rainfall time-series are used in this study for the 1960–2016 period. They were obtained from continuous measurements at the synoptic weather station of Conakry in Guinea. A set of 684 monthly average temperature measurement during 57 years were used, for rainfall, the same data number were used too. The location of this synoptic station at the international airport of Conakry makes the data set uninterrupted and of good quality. The daily mean temperatures were calculated by averaging the daily minimum and maximum temperatures. The monthly and yearly temperature averages were calculated from the daily and monthly averages, respectively, for the complete study period. The histogram of monthly mean temperature peaks at 26 °C with 133 occurrences (Figure 2a).

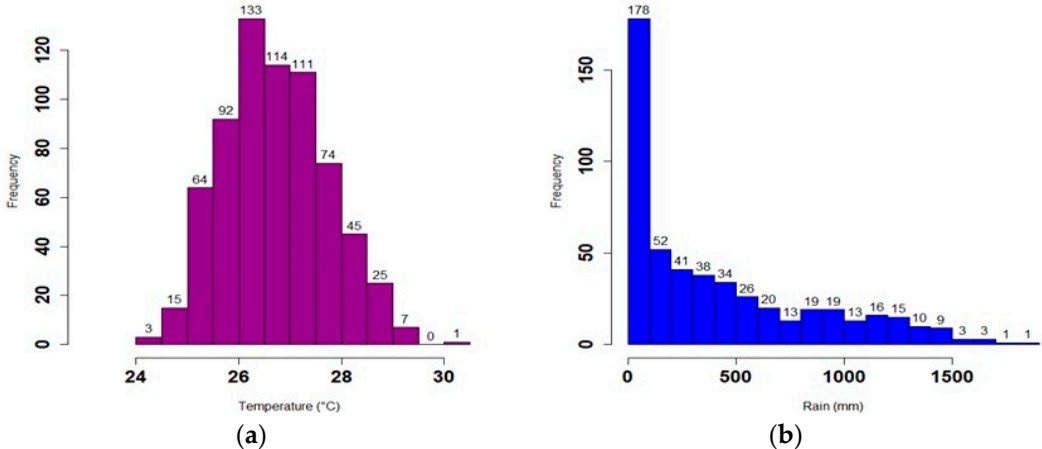

**Figure 2.** Histograms of monthly temperature frequency, the monthly temperature mean value of 26 °C has higher frequency of observation (**a**) and monthly rainfall (**b**) overall frequency showing that the monthly rainfall value of 0.1–100 mm has higher frequency of observation at Conakry Airport station.

The monthly rainfall is the accumulated based on daily rainfall obtained for a particular month. The overall annual rainfall is calculated as the sum of monthly rainfall. However, measured rainfall commonly consists of discrete series of rainfall events with different durations and time intervals [23]. It is noteworthy that rainfall is a discontinuous parameter, thus, monthly rainfall accumulated values used in our study oscillate between 0.1 and 1839.3 mm. Rstudio and Matlab software were used to perform all computational tasks. The histogram of monthly accumulated rainfall peaks at 0.1–100 mm with 178 occurrences, such as values above 1500 mm, have a lower occurrence (<10), but are very quantitatively significant from disaster (flood, landslide) point of view (Figure 2b).

To achieve a better understanding of the forcing that may influence temperature and rainfall of Conakry, four climate indices were used:

1.　Niño3.4 monthly mean time series from 1960 to 2016 (684 measurements) were downloaded from the National Oceanic and Atmospheric Administration (NOAA) website (https://www.esrl. noaa.gov/psd_wgsp/Timeseries). The Niño3.4 index is calculated by taking the area-averaged sea-surface temperature (SST) within the Niño3.4 region, which extends from 5° N to 5° S in latitude and from 120° W to 170° W in longitude (in the Pacific Ocean). We use Niño3.4 averages calculated from the HadISST SST dataset, which is given by 1° in latitude–longitude.

2.　Atlantic Meridional Mode (AMM) SST index from 1960 to 2016 (684 data) were downloaded from the National Oceanic and Atmospheric Administration (NOAA) website (https://www.esrl.noaa. gov/psd/data/timeseries/monthly). The AMM time series is calculated by projecting SST on to the spatial structure resulting from Maximum Covariance Analysis (MCA) to sea surface temperature (SST) over the region of 21° S–32° N, 74° W–15° E.

3.　Tropical Northern Atlantic index (TNA) is the anomaly of the monthly averaged SST values. The TNA SST index is defined as region-averaged SST anomalies in the domain (0°–20° N, 60° W–20° E) [24]. TNA monthly mean time series from 1960 to 2016 (684 data) were downloaded from the NOAA website (https://www.esrl.noaa.gov/psd/data/climateindices/list/#TNA).

4.　The Atlantic Niño (AN) or Atlantic Equatorial Mode is a quasiperiodic interannual climate pattern of the equatorial Atlantic Ocean. The term Atlantic Niño comes from its close similarity with the El Niño-Southern Oscillation (ENSO) that dominates the tropical Pacific basin [25]. The Atlantic Niño (AN) index is defined as the SST anomaly in the central-eastern tropical Atlantic: (3° S–3° N; 20° W–0° E) [26]. The AN monthly mean time series from 1960 to 2016 (684 measurements) were download from KNMI website (https://climexp.knmi.nl/start.cgi). The equatorial warming and cooling events associated with the Atlantic Niño are known to be strongly related to atmospheric climate anomalies, especially in African countries bordering the Gulf of Guinea [27]. As Conakry is a coastal region in west Africa, we used AN in our study.

In this work, the monthly average temperature (684 measurements) and monthly rainfall (684 measurements) as well as the monthly mean (684 measurements) of 4 climatic indices (Nñio 3.4, AMM, TNA and AN) were used as input data for our investigation during the period 1960–2016.

*2.2. Method*

According to the World Meteorological Organization [28,29] (WMO), climatological standard normals are defined as the averages calculated for uniform and relatively long period including at least three consecutive periods of ten years. The climatological standard normals are hence the averages of the climatological data calculated for the consecutive periods of 30 years. Our climatological study is based on the standard normal calculated for the 1961–1990 period. The obtained arithmetic mean was calculated (for temperature and rainfall) by using the following formula:

$$X = \frac{1}{n} \sum_{i=1}^{1} (x_i), \tag{1}$$

where, $n$ = year number.

Over the Conakry site, the climatological normal calculated for temperature and rainfall on the 1961–1990 period corresponds to the values of 26.5 °C and 3806.8 mm, respectively. These values are used to calculate corresponding anomalies.

### 2.2.1. Mann-Kendal Test

It is always essential to work out monotonic trends in the time series of any geophysical data before any further use. In this study, the Mann-Kendall test [30–32] was employed to detect the trends that exist in both the temperature and rainfall time series. This method is defined as a non-parametric, rank-based method which is commonly used to extract monotonic trends in the time series of climate data, environmental data or hydrological data. The Mann-Kendall test statistic gives information about the trend of the total time series and its significance. However, it is important to investigate how the trend varies with respect to time. Therefore, the calculation of the forward/progressive ($u(t)$) and backward/retrograde ($u'(t)$) values of the Mann-Kendal test statistic is essential in order to investigate both the potential trend turning points and the general variability of trends in respect to time. This method is called the Sequential Mann-Kendall (SQ-MK), and it is well explained by [33] and other authors [31,34]. This method has been found to perform very well in trend analyses of stream flow and precipitation [35] and also in the field of earth remote sensing [36].

### 2.2.2. Multilinear Regression

One of the primaries aims of this study is to identify the relationship between the studied time-series (temperature and rainfall at Conakry station in the present paper) and climate indices such as TNA, Niño3.4, AMM and AN. The multi-linear regression (MLR) is a method that is frequently used to explain the relationship between one continuous dependent and two or more independent variables (climatic indexes in this case). The MLR model output $y_i$ based on a number "$n$" of observations can be expressed as follow:

$$y_i = \beta_0 + \beta_1 x_{i2} + \cdots + \beta_p x_{ip} + \varepsilon_i, \tag{2}$$

where

$$i = 1, 2, 3, \ldots, n, \tag{3}$$

where in this case, $y_i$ is the dependent variable $x_{ip}$ represents the independent variables, $\beta_0$ is the intercept, and $\beta_1, \beta_2, \ldots \beta_p$ are the $x$'s coefficients. The final term ($\varepsilon_i$) represents the residual term which the model should always keep its contribution as minimum as possible.

### 2.2.3. Theil-Sen's Estimator

Theil-Sen slope estimate method were used to analyse the long-term trend in the data and the seasonality. The Theil-Sen estimator (TSE) is fairly resistant to outliers and is robust with a high breakdown point of 29.3% [37,38]. TSE method was first outlined by Theil [39] and later expanded upon by Sen [40]. The determination of trend slope of n-pair of data is given by the formula:

$$T_i = \frac{x_j - x_i}{j - i}, \tag{4}$$

where, $x_j$ and $x_i$ presents as data values at time $j$ and $i$ ($j > i$) respectively [37].

### 2.2.4. Wavelet Analysis

The present study employed the Morlet wavelet which provides a good balance between time and frequency localization [41], especially for geophysical data. Wavelet analysis includes different wavelet functions such as the windowed Fourier transform, wavelet transform, normalization, wavelet power spectrum, etc. The main advantage of the wavelet analysis in comparison with other techniques is that it analyses localized variations of power within a time series. By decomposing a time-series into

time-frequency space, one is able to determine the dominant modes of variability and their variation with time [42]. Wavelet transform coherence (WTC) is a good method for analyzing the coherence and phase lag between two time-series as a function of both time and frequency [43]. Therefore, we adopted the Monte Carlo wavelet and coherence analysis to quantify the relationships between climate forcing and the two data sets (rainfall and temperature) recorded at Conakry. More details about wavelets and wavelet coherence and phase are given by Torrence and Compo [42], Grinsted et al. [41] and others.

Basically, from a climatological point of view, the 1961–1990 normal (30 years) was used for this study. The models used thus show complementarity in the sense that the Mann-Kendall test gives information about the trend of the total time series and its significance. In addition, SQ-MK is important to determining both the trend variability in time and the trend change points in the time series. However, it is important to identify the relationship between the studied time-series and climate indices, for that purpose, the MLR and Wavelet are used. But the difference between these two models is that the multi-linear regression (MLR) helps to explain the relationship between one continuous dependent and two or more independent variables. The Wavelet analysis method helps to determine the dominant modes of variability and their variation with time, in addition it helps to quantify the relationships between climate forcings and the two data sets indicating the period when the correlation is significant as well. Furthermore, it also specifies whether the parameters are correlated or not and if so, whether they are in-phase or out-of-phase or if the causal relationship is identified or if there is simultaneity. The results from this methodology are then discussed in the following sections and some figures are plotted according to that done by Bilbao et al. [44].

## 3. Results

### 3.1. Climatology and Seasonality of Temperature and Rainfall

3.1.1. Inter-Annual Variation of Temperature and Rainfall

Figure 3a shows the month versus year evolution of the monthly averaged temperatures recorded at Conakry station from 1960 to 2016. This figure indicates that Conakry is experiencing an increasing temperature, found to be significant since 1970s. It is clearly shown on this figure that 1998 is the year with the highest temperature (30 °C in mean recorded on April). The 1997/98 El Niño phenomenon, which started in March 1997 and lasted until mid-1998, had resulted in severe flooding and drought in several parts of the world [45,46].

By analysing the evolution of the temperature in two different periods, 1960–1998 and 1998–2016 (Figure 3c), we have found an increase of 0.8 °C from the first period to the 2nd one. In average, the temperature ranges from 26.5 °C to 27.8 °C from one period to the other. The annual averaged temperature of 1998 is 28.1 °C. Similar analysis for another Guinean station located at 7.74° N; 8.82° W, 900 km far from Conakry is reported by Loua et al. [9]. They highlighted a warming due to the increase in evaporation. We assume that the 1998 warming observed at Conakry seem to be linked to the 1998's strong El Niño. Angell et al. [47], shown that the record global warmth in 1998, particularly in the 850–300 mb layer, is partly, if not mostly, due to the very strong El Niño of 1997–1998. Strong El Niño event made 1998 relatively hot at the surface and in the atmosphere. The exceptionally warm El Niño year of 1998 was an outlier from the continuing temperature trend. Previous works have also pointed out the influence of the large tropical explosive volcanic eruptions and ENSO on precipitation and temperature changes over West Africa [48–50]. However, these studies reveal that thus far no consensus has been reached on either the sign or physical mechanism of El Niño response to volcanism. Based on the use of the Fifth Coupled Model Intercomparison (CMIP5), Khodori. [49] showed that large tropical volcanic explosions favour an El Niño within 2 years following the eruption. They demonstrated that volcanically induced cooling in tropical Africa weakens the West African monsoon and the resulting atmospheric Kelvin wave drives equatorial westerly wind anomalies over the western Pacific. This wind anomaly is further amplified by air–sea interactions in the Pacific, favouring an El Niño-like response. This analysis was found in agreement with the study reported by

Liu et al. [50]. Through the use of the Community Earth System Model (CESM1), they shown that volcanic eruptions are efficient in reducing the monsoon precipitation. In addition to reduce moisture heavily, the volcanic eruptions can affect the circulation field much [50].

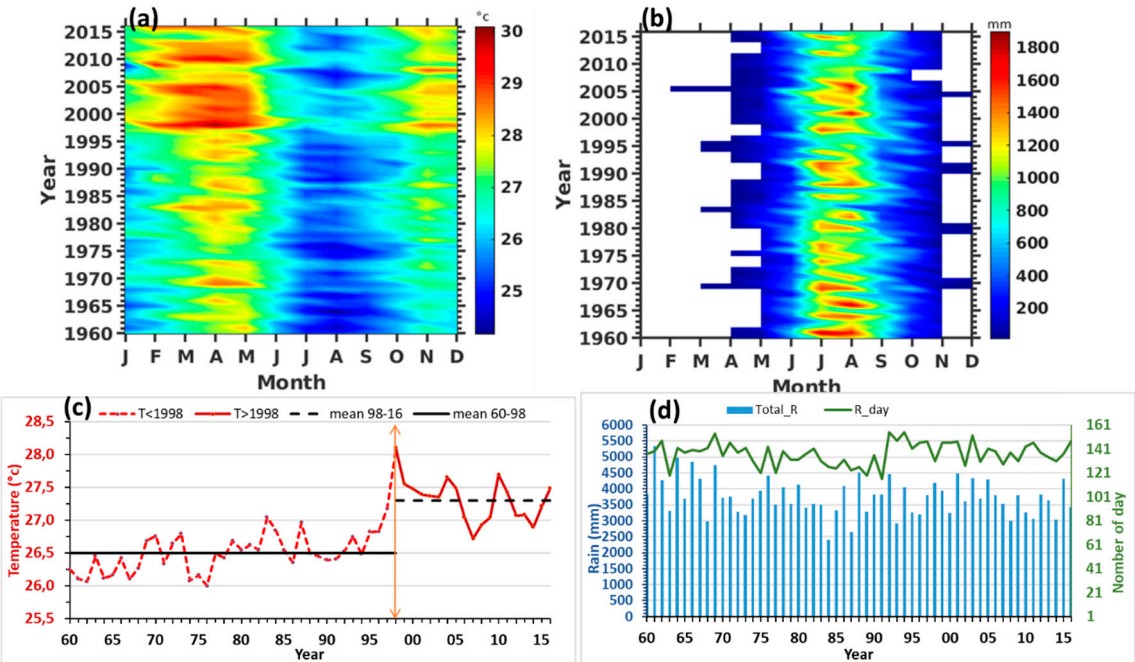

**Figure 3.** Yearly/monthly evolution of temperature (**a**) and rainfall with a pick during June-July-August (**b**); interannual evolution of temperature, dotted vertical line is the year 1998 (**c**) and interannual evolution of rainfall (blue bars) with rain day (black solid line) (**d**).

For the 1998 warming, Wang Shaowu et al. [51] explained that it is evident the annual temperature of 1998 set the highest record for the past century in China. Foster and Rahmstorf [52] reported the strong influence of known forcings on short-term variations in global temperature, including El Niño–Southern Oscillation (ENSO), and to a lesser degree, solar cycles. It so happens that 1982–1983 and 1997–1998 were the times of two biggest El Niño on record, and it is well established that a mini global warming occurs at the latter stages of an El Niño as heat comes out of the upper ocean and contributes to a warmer atmosphere and surface, but resulting in a cooler ocean [53].

The Figure 3b shows the yearly/monthly evolution of rainfall. There are climate conditions where one summer may be sunny, dry, and warm, whereas another may be cool, cloudy, and wet. Globally, the biggest cause of such regimes that last several seasons is the ENSO phenomenon [54]. For the specific case of Conakry, we remark that the evolution of temperature and rainfall through Hovmöller representation (year/month) shows an interseasonality for rainfall, and an increase in temperature. Additionally, for each year, the monthly maximum values of rainfall (>1500 mm) are recorded during the period June-July-August-September. Smallest amount of annual overall rainfall was recorded on 1984 (Figure 3d) and seem to be linked to the 1980s severe drought [55,56] that West Africa has experienced. On the one hand, the increase in temperature observed may be a response to global warming, and it is therefore consistent to diagnose whether warm years coincide with the occurrence of some geophysical phenomena such as El Niño. On the other hand, the remarkable interseasonality can be associated not only to the climatic warming but also to the irregularity in the intensification of the WAM and the dynamics of the ITCZ.

From this general overview of the interannual evolution of these meteorological parameters, we therefore proceeded to the analysis of the monthly climatology in the following section. This analysis allowed us to better understand the variability of the monthly climatological average of each parameter

during the year. It will therefore be necessary to highlight the different seasons to which this region is subject.

### 3.1.2. Monthly Climatological Variations

Figure 4a depicts the variation of monthly climatology of temperature during the year over the whole period of observation at Conakry. Climatologically, the variation of monthly mean temperature shows clearly that the semi-annual cycle is dominant than the annual cycle. During the year, the temperature means oscillate between 24.2 °C and 30.1 °C with an annual mean of 26.8 °C. During the winter season, the monthly mean temperature may reach a peak in November (27.3 °C) and a second one in April (28.1 °C). In summer (June–October), it decreases in August (25.4 °C). This abrupt decrease in temperature that starts in may corresponds to the beginning of the rainy season, remarkably, a strong shift appears in June which seems to be due to the onset of WAM. Sylla et al. [57] reported that the beginning of rainy season in the West Africa Region can be associated with the northward migration of ITCZ from 4° N to 10° N, and the onset of the West African summer monsoon in the second half of June.

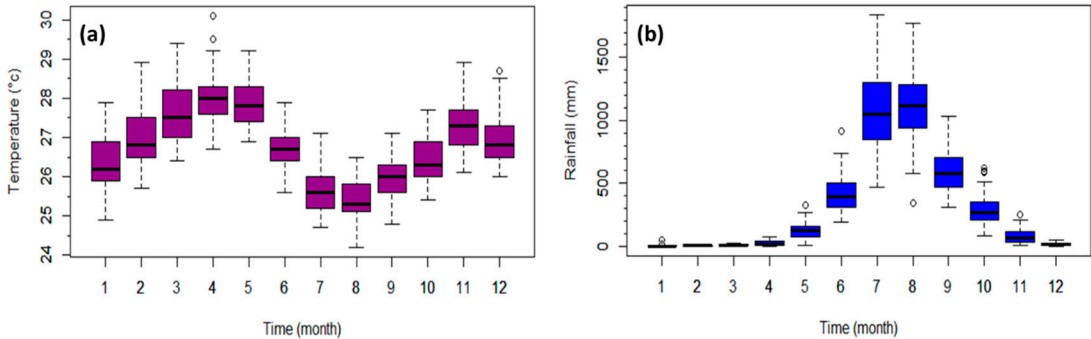

**Figure 4.** Climatology of monthly temperature showing two picks on April and November (**a**) and monthly rainfall showing a pick on July (**b**) as derived from ground observations at Conakry station from 1960 to 2016.

While during summer, under a cloudy sky, or overcast and rainiest, there is less solar radiation that reaches the Earth's surface. The temperature remains relatively low, resulting in a small thermal amplitude. The equatorial cooling intensifies the southerly monsoon in the Gulf of Guinea and pushes the continental rain band inland from the Guinean coast [11].

The Figure 4b shows the evolution of monthly climatology of rainfall at Conakry during the year. The variability of rainfall during the year shows an annual cycle with a peak recorded on August (>1000 mm). It is clear that the rainfall becomes significant in May, that corresponds to the beginning of summer (ICTZ northward migration), and it is followed by an abrupt upward jump in June (WAM onset) before reaching the peak in August (ICTZ at 10° N). During that period, the temperature decreases gradually from April to reach the minimum in August (Figure 4a). By the beginning of September, the rainfall is characterized by abrupt downward jump when the temperature starts increasing (ICTZ downward migration and weakening of WAM), and then the latest rains in the year are recorded in November. The beginning and end of rainy season are characterized by high frequency of strong storms in Guinea [9].

### 3.1.3. Temperature and Rainfall Anomalies

Temperature/rainfall anomaly from normal calculated for the period from 1961 to 1990 refers to the difference in degrees Celsius/in millimeter between the average annual temperature/annual rainfall observed from 1960 to 2016 in comparison with the average annual temperature/annual rainfall observed during the period from 1961 to 1990.

In this study, annual averaged temperatures were standardized by using the average of the period 1961–1990 (26.5 °C). In Figure 5a, the blue (red) bars indicate the negative (positive) anomalies and the fit line shows upward trend of temperature. Temperature anomalies could be classified in three classes:

- (a) the cold class: it corresponds to periods with negative anomalies (1960–1962; 1964–1965; 1967–1968; 1971; 1974–1976; 1986);
- (b) the quiet or normal class with temperature anomalies close to zero (1963; 1966; 1977–1978; 1985–1986; 1988–1992, 1994) and;
- (c) the warm class with positive anomalies (1969–1970; 1972–1973; 1979–1984; 1987; 1993; 1995–2006). The last period of the warm class is the warmest and longest one, it lasts about 12 years (1995–2006).

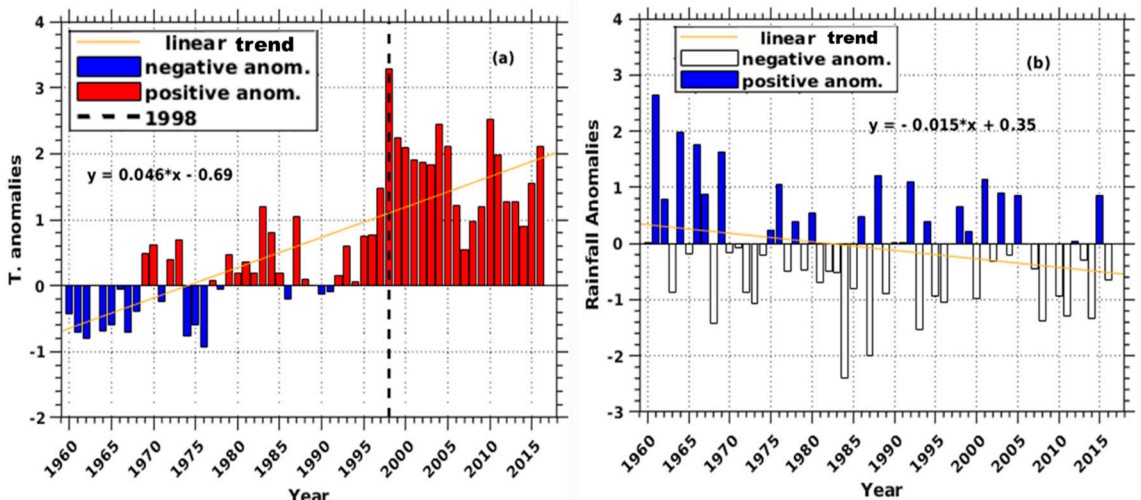

**Figure 5.** Temperature anomalies with dotted vertical line showing the year 1998, the blue (red) bars are negative (positive) anomalies and the increasing linear trend (**a**). Rainfall anomalies with decreasing linear trend, blue (white) bars indicate positive (negative) anomalies (**b**) of Conakry airport station: 1960–2016.

These results are consistent with those reported by Loua et al. [9]. On the whole, the inter-annual evolution of temperature shows a predominance of the warm class (positive anomalies) since 1992, and then all the following years are classified as warm, with a maximum in temperature anomalies obtained in 1998 (higher than + 3 °C). Among the most intense El Niño episodes of the last forty years, the one of 97–98 was the one that triggered the earliest and most severe. The countries most affected in terms of their infrastructure were USA, Indonesia and Brazil, but the highest human losses remain for Africa [58]. To confirm our result by the global analysis of surface temperature, Simmons et al. [59] highlighted that surface warming from 1998 to 2012 is larger than indicated by earlier versions of the conventional datasets used to characterize what the fifth assessment Report of the Intergovernmental Panel on Climate Change (IPCC) termed a hiatus in global warming.

Figure 5b illustrates the rainfall anomalies corresponding to the period from the year 1960 to 2016, using 1961–199 standard normal (3806.8 mm). It shows that there are both positive (blue) and negative (magenta) anomalies of rainfall during the study period. The positive (negative) anomalies correspond to wet (dry) years, and consecutive years (1970–1974) and (1981–1985) define two driest periods which correspond to two drought events in west Africa (1970s and 1980s). Peel et al. [60] highlighted that the consecutive dry years are associated with drought, which is a significant physical and economic phenomenon that imposes great stress on ecosystems and societies. However, drought is a part of natural climatic variability on the African continent, which is high at intra-annual, inter-annual, decadal and century timescales [61]. Where considering both the temperature and rainfall anomalies (Figure 5), we may notice that during the years 1970 and 1980 there were severe drought episodes in the study

area. This was also reported by previous studies such as [62,63]. The West African Sahel is well known for the severe droughts that ravaged the region in the 1970s and 1980s [17].

This section allowed us to identify periods of hot consecutive years (70, 84 and 1992–2016) as well as periods of consecutive dry years (70–74 and 81–85). These periods served us as important references to take into account for the rest of the analysis on the variability of the trend and with the forcings used as well.

### 3.2. Trend Analysis of Temperature and Rainfall

In this study, the Mann–Kendall (MK) trend test was used. In regards to the temperature time series, a significant positive Z-scores value (9.3067) which is far greater than 1.96 was found, suggesting that the temperature trend is increasing. However, for rainfall, the MK trend test shows no-significant negative Z score (−0.17143) which above −1.96, suggesting weak decrease in rainfall variability.

Figure 6a shows the sequential statistic values of forward/progressive (Prog) u(t) (solid red line) and retrograde (Retr) u'(t) (black solid line) obtained by SQ-MK test for Conakry yearly mean temperature. In general, SQ-MK indicates and upwards trends of temperature in Conakry which is noticeable in both Prog. and Retr. SQ-MK statistic. The possibility is that the upwards trend started before the beginning of the time series (1960) because the change detection point, a point where Prog. and Retr. cross each other did not occur in the graph. What is noticeable in this figure is that, it is only from 1984 that this progressive SQ-MK statistic becomes positive and significant. At the same pace, it gradually increases until 1989 and then stands until 1998, the year from which the trend has increased significantly far above the confidence level (+3.866541) up to 2016 (+9.306717). There is a significant upward trend which seems to coincide with the 1970s 1980s droughts episodes and strongly the 1998 and 2014–2016 strong El Niño event. In a study that uses the similar non-parametric test method, Suhaila et al. [64] reported that the detection points captured by Pettitt and SQ–MK tests in Peninsular Malaysia temperature series during the years 1996, 1997 and 1998 are possibly related to climatic factors, such as El Niño and La Niña events. The retrograde statistic values are significant and negatives during the period from January 1960 to 1992 before it continues to be within the 95% confidence level limits (±1.96) except the year 1998 which the retrograde statistic value is significantly positive.

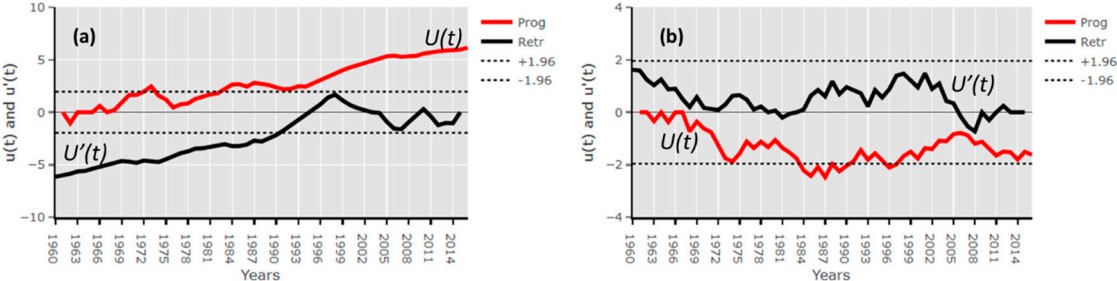

**Figure 6.** Sequential Mann-Kendal statistic values of progressive u(t) (solid redline) and retrograde u'(t) (black solid line), obtained by Sequential Mann-Kendall test for temperature (**a**) and rainfall (**b**) of Conakry airport: 1960–2016.

Figure 6b depicts the sequential statistic values of forward/progressive (Prog) u(t) (solid red line) and retrograde (Retr) u'(t) (black solid line) obtained by SQ-MK test for Conakry annual rainfall data for the period from 1960 to 2016. A strong significant upward trend was observed in late 1961, with the significant trend turning point observed in June 1962, which means that 1961 is the only year that is characterized by a positive and significant trend over the entire study period. But a careful analysis of the trend in progressive and retrograde which are non-significant (between ±1.96) and sometimes negative or positive shows two distinct periods that correspond to that found by the analysis of precipitation anomalies. For the first period (1970–1974) and the second period (1981–1985), the Retrograde curve is below the progressive curve in the negative band, which corresponds to

periods of deficit rainfall. For the rest of the study period, the two curves intersect each other or the retrograde curve is above the progressive curve, that corresponds to periods with variable or normal rainfall. The response of the West African drought of 1970s and 1980s is clearly identified by the reduction in the rainfall at Conakry. Statistically there is a no-significant downward trend in rainfall since the end of 1960s.

In summary, the SQ-MK test and MK model for Conakry yearly data shows that the temperature and rainfall are subject to a significant increasing trend and a no-significant decreasing trend, respectively, during the period from 1960 to 2016. Thus, these methods seem to be useful for explaining the variability and trends of both temperature and rainfall.

In order to investigate physical relationships between climate forcing, precipitation and streamflow in the Mid-Atlantic region, Schulte et al. [4] selected eight climate indices. In the present study, four climate indices (Niño3.4, AMM, TNA and AN) were used as explanatory variables for this model because of their well-known possible influence on temperature and rainfall variability over West African region. The relevant time series of these climate indices are shown in Figure 7. Zebiak [27] specified that the dominant signature of ENSO is clearly focused on the Equator and its temporal variability is strongly focused at 3–5-year time scales.

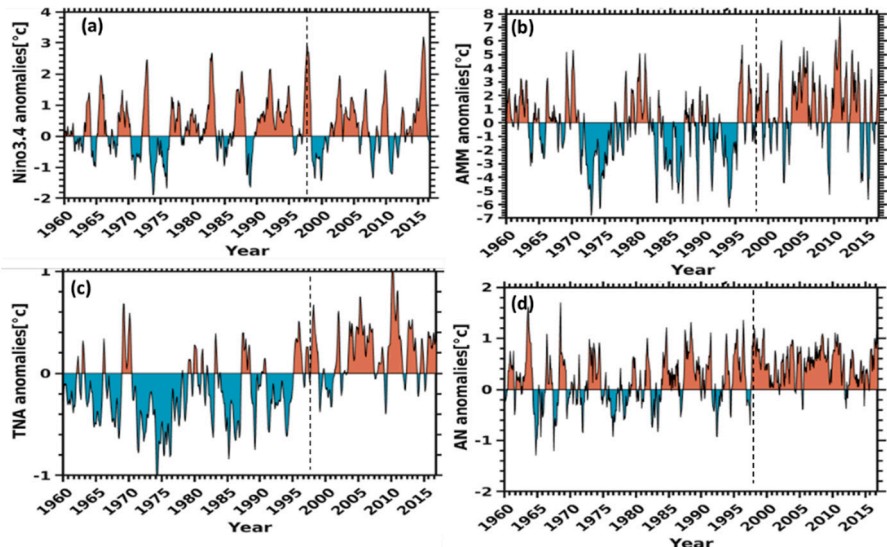

**Figure 7.** The standardized monthly Niño3.4 (**a**), AMM (**b**), TNA (**c**) and AN (**d**) time series for the period from 1960 to 2016. The vertical dashed lines indicate the year 1998.

There are two main forms of coupled ocean–atmosphere variability that exist in the tropical Atlantic Ocean, namely: the first one Atlantic Meridional Mode (AMM) [65] which is also called the interhemispheric mode [66]. It was originally identified by Servain [67]. This mode of variability is characterized by an interhemispheric gradient in sea surface temperatures and by oscillations in the strength of surface winds that cross the Equator, thereby reinforcing sea surface temperature anomalies [68]. The pronounced coupled ocean-atmosphere variability in the Tropical Atlantic is generated by fluctuations in the Atlantic Meridional Mode (AMM) [68]. The AMM is characterized by an anomalous meridional shift in the Intertropical Convergence Zone (ITCZ) that is caused by a warming (cooling) of SSTs and a weakening (strengthening) of the easterly trade winds in the northern (southern) tropical Atlantic [69]. And, the second one is the zonal mode, also called the Atlantic Niño [70]. Its seasonal evolution is due to surface wind variations associated with the northward migration of the ITCZ [71].

The tropical northern Atlantic (TNA) SST anomaly pattern is an important component of the tropical Atlantic SST variability, which is characterized by warm (or cold) SST anomalies in the TNA [72]. Sea surface temperatures in the tropical North Atlantic (TNA) affect the meridional movement of the

ITCZ and its band of heavy rainfall and cloud cover [73]. The Atlantic Niño (AN) is often regarded as something like the little brother of El Niño. During Pacific El Niño events, sea-surface temperatures (SSTs) in the central and eastern equatorial Pacific become warmer than average. Prevailing theories on the equatorial Atlantic Niño are based on the dynamical interaction between atmosphere and ocean [74]. In very much the same manner, SSTs in the central and eastern equatorial Atlantic become warmer than average (or anomalously warm) during Atlantic Niño events. The Atlantic Niño index used in this study is obtained by calculating the area average of SST in the cold tongue region, defined as 20° W to 0 and 3° S to 3° N [26]. While El Niño usually peaks in northern hemisphere winter, the Atlantic Niño peaks in summer [75]. Therefore, understanding of the Atlantic Niño (or lack thereof) has important implications for climate prediction in those regions. Although the Atlantic Niño is an intrinsic mode to the equatorial Atlantic [27].

There may be a tenuous causal relationship between climate parameters and the Atlantic Niño in some circumstances. Therefore, MLR and Wavelet analysis are used to identify the dependency and coherence between temperature, rainfall and climate forcings.

The correlations between the four indices used in our study are shown in Figure 8. There is strong correlation coefficient between AMM and TNA (0.76). As the two explanatory variables are strongly correlated, the MLR analysis may have difficulties to separate the contributions. For that purpose, the wavelet analysis was used by calculating the coherence between explanatory variables and the dependent variable separately.

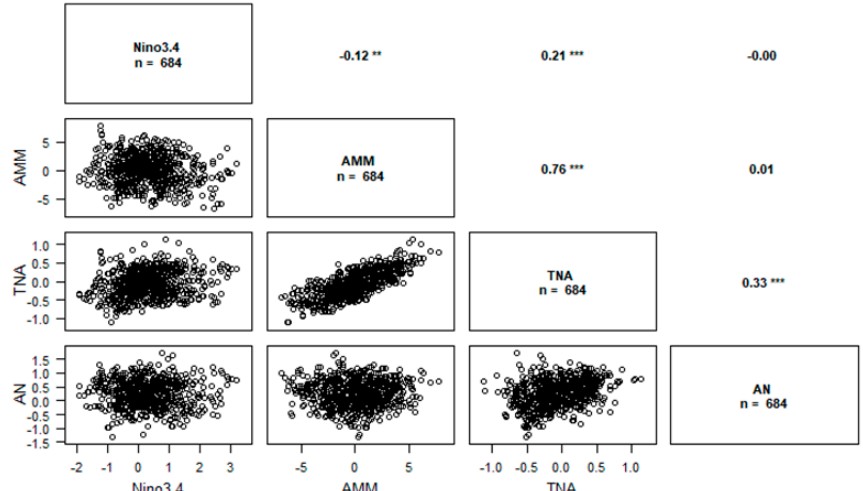

**Figure 8.** Correlation between the standardized monthly Niño3.4, AMM, TNA and AN time series for the period from 1960 to 2016, "n" is data number. The values are the correlation coefficient between the 4 parameters and we found that TNA and AMM are significantly correlated. The "***"; "**"; "*"; show that the correlation is significant to the 0.001; 0.01; 0.1 level and " " mean that there is no correlation.

For the MLR the explanatory variables (AMM + Niño3.4+ TNA + AN) were used. The output of the MLR statistical analysis of temperature and the independent variables is shown in Table 1. Statistically the results in Table 1 reveal a significant relationship between temperature and Niño3.4 AMM, and TNA, with *p*-values of 0.0138, $1.99 \times 10^7$, and less than $2 \times 10^{16}$, respectively. The *p*-value for AN indicates a statistically insignificant association with the temperature because of *p*-value which is far greater than 0.05.

**Table 1.** The output of Multiple Linear Regression (MLR) model in which temperature is a dependent variable and AMM, Niño3.4, TNA and AN are independent variables.

| Variables | Estimate | Std. Error | *t*-Value | *p*-Value | Significance |
|---|---|---|---|---|---|
| **Niño 3.4** | −0.11653 | 0.04722 | −2.468 | 0.0138 | * |
| **AMM** | −0.14055 | 0.02675 | −5.254 | $1.99 \times 10^{7}$ | *** |
| **TNA** | 1.79296 | 0.20497 | 8.747 | $2 \times 10^{16}$ | *** |
| **AN** | 0.14372 | 0.08790 | 1.635 | 0.1025 | |

Significant codes: 0 '***' 0.001 '**' 0.01 '*' 0.05 '.' 0.1 ' ' 1.

A comprehensive summary of the MLR analysis statistics encompassing rainfall, Niño3.4, AMM, TNA and AN is shown in Table 2. The results in Table 2 reveal a statistically significant relationship between rainfall and Niño3.4, AMM, and TNA with *p*-values of 0.04374, 0.00441, 0.00301. The *p*-value for AN indicates a statistically insignificant association with the rainfall because of *p*-value (0.73691) which is far greater than 0.05. A strong dependence between the two meteorological parameters, AMM and TNA were found. And then, Niño3.4 has a moderate influence on the temperature and rainfall of Conakry. The low dependency between the AN and these two meteorological parameters would be due to the distance between the Conakry site and the NA focus (3° S–3° N), and this could be verified for another station closer to the equator. To analyse the two-component dependence of which one in the temporal environment and the other in the frequency environment between the temperature, the rainfall and the forcings used in the study, unlike the MLR, the wavelet model has been evaluated and model outputs are explained in the next section.

**Table 2.** The output of Multiple Linear Regression (MLR) model in which rainfall is a dependent variable and AMM, Niño3.4, TNA and AN are independent variables.

| Variables | Estimate | Std. Error | *t*-Value | *p*-Value | Significance |
|---|---|---|---|---|---|
| **Niño 3.4** | 43.55 | 21.55 | 2.020 | 0.04374 | * |
| **AMM** | 34.89 | 12.21 | 2.857 | 0.00441 | ** |
| **TNA** | −278.57 | 93.56 | −2.977 | 0.00301 | ** |
| **AN** | −13.49 | 40.12 | −0.336 | 0.73691 | |

Significant codes: 0 '***' 0.001 '**' 0.01 '*' 0.05 '.' 0.1 ' ' 1.

An influential variable for most African rainfall areas is the zonal wind over the tropical Atlantic, the north-south SST gradient in the tropical Atlantic modulates rainfall in West Africa as expected [76].

Figure 9a depicts the time evolution of mean monthly temperatures, with the warming trend line superimposed. From Theil-Sen function, in this study, 684 points were used for trend estimation. The trend estimate is: $p < 0.001 = ***, p < 0.01 = **, p < 0.05 = *$ and $p <0.1 = +$. The temperature increases at 0.02 °C per year (0.2 °C/decade) at Conakry. The superimposed red line indicates the obtained linear trend. And the dashed red lines indicate the 95% confidence interval. The annual evolution of rainfall exhibits a negative slope, which corresponds to decreasing trend (Figure 9b) at −8.14 mm per year (−81,4 mm/decade). Compared to other sites in West Africa, our results are similar to that found by [32], for stations in downstream Kaduna River Basin during 1975–2014, in Nigeria. The fifth Intergovernmental Panel on Climate Change assessment stated Africa surface temperature already increased by 0.5 °C–2 °C over the past hundred years and an observed drop in average annual rainfall of approximate 25–50 mm each decade from 1951–2010 in some parts of West Africa [77]. Globally, according to the IPCC Special Report [78], it has been reported that the warming of anthropogenic origin has already exceeded the environment.

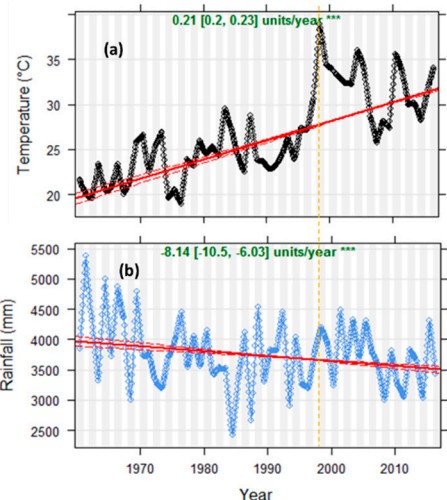

**Figure 9.** Shows the long-term trend of monthly temperature (**a**) and rainfall (**b**). The solid red line shows the trend estimate and the dashed red lines show the 95% confidence intervals for the trend based on resampling methods. The overall trend is shown at the top-left as 0.21 °C per year (**a**) and −8.14 mm per year (**b**), and the 95% confidence intervals in the slope from 0.2–0.23 °C/year (a) and −10.5–6.03 mm/year (**b**). On the figures, the sign "***" shows that the trend is significant to the 0.001 level.

The seasonal distribution of temperature is shown in Figure 10a. the increase in temperature is more significant in winter (December–January–February) of 0.03 °C per year than spring (March–April–May), summer (June–July–August) and autumn (September–October–November) of 0.02 °C per year. The Figure 10b depicts the seasonal distribution of rainfall; no significant trend is observed in the winter months and rainfall values seem to be stables. Negative linear trend was found in spring (−0.34 mm/year), autumn (−1.23 mm/year) and for summer, the trend is positive (0.1 mm/year).

### 3.3. Wavelet Analysis

The Figure 11 shows the normalised wavelet power spectrums calculated for the time series of temperature (a) and rainfall (b) for the period from year 1960 to 2016. In this figure, the "u" shaped solid lines represent the cone of influence (COI) which define the region of the spectrum which should be considered in the analyses. The COI actually indicates areas where edge effects occurs in the time series [44]. The thick black contours are the 95% significant regions of confidence level [42].

The main purpose of using the wavelet transform technique in our study is to identify any dominant variability mode that may be present within the two meteorological parameters (temperature and rainfall). In general, the wavelet transform of temperature shows two distinctive peaks (Figure 11a) corresponding at 6 and 12-month periods. The 12-month period shows strong power spectrums during the distinctive periods of 1961–1965, 1970, 1984 and 1992–2016. Its intensity increases with no-interruption from 1992 to 2016. These dominant wavelet peaks seem to be consistent with the results presented in Figure 3c. The wavelet power spectra for rainfall indicate a strong power spectral peak of 12-month cycle which starts from year 1960 to 2016 (Figure 11b). Moreover, there is a weak power that seems to appear at 6-month period for a few years and within the 95% significant regions of confidence level for distinctive years 1961, 1965, 1967, 1970, 1986–1992, 1998–2000 and 2002–2007.

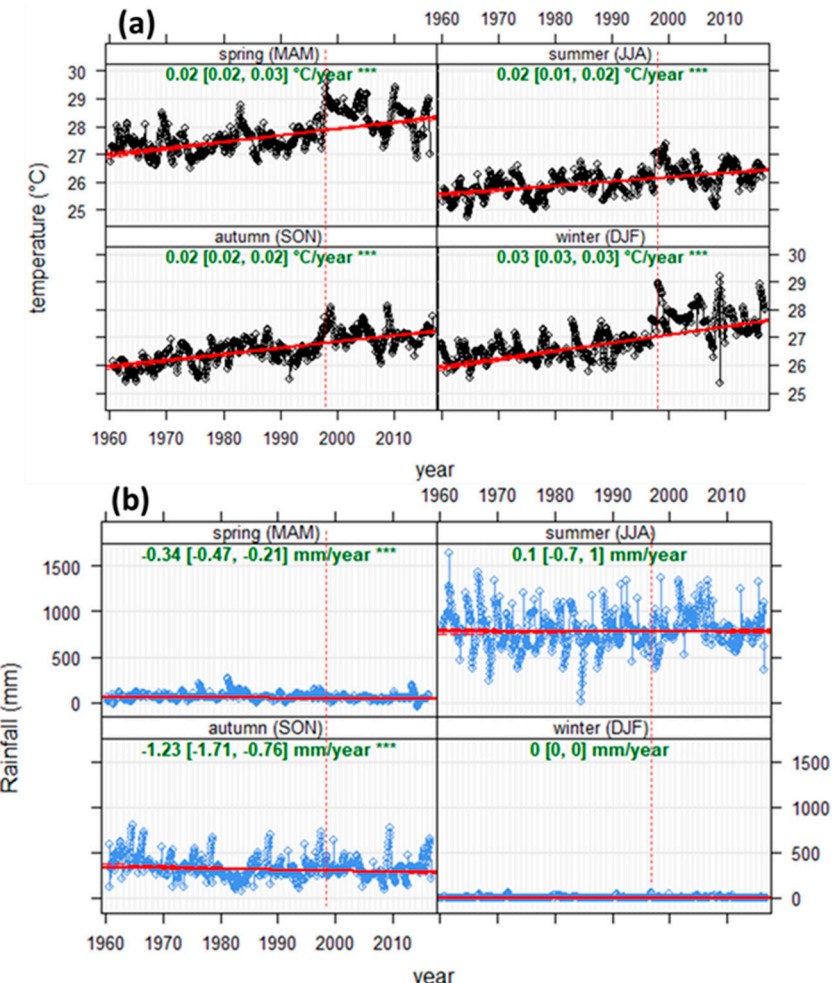

**Figure 10.** The plot shows the Seasonal trend distribution of monthly mean temperature (**a**) and rainfall (**b**) obtained with Theil-Sen's estimate. The solid red line shows the trend estimate and the dashed red lines show the 95% confidence intervals for the trend based on resampling methods. The vertical red line indicate year 1998. There is no trend in rainfall during winter (0 mm per year).

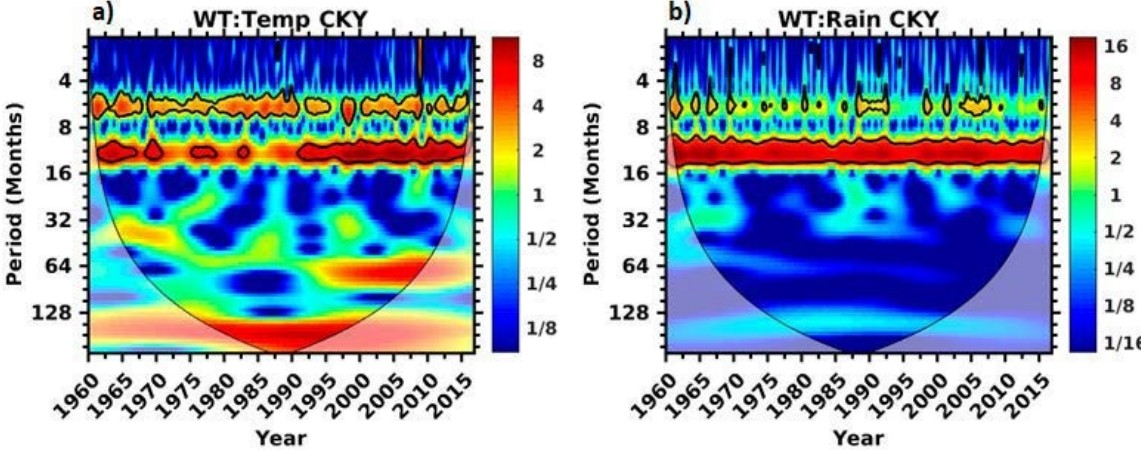

**Figure 11.** Wavelet transform of temperature (**a**) and rainfall (**b**) variability from 1960 to 2016 at Conakry. The black solid line contour delimits the region (red) where the power is strong and significant and the cone of influence indicates the 95% confidence level.

It is important to note that the distinctive power periodicities found in temperature may be associated with the annual and semi-annual cycles, which are controlled by the alternation between dry and wet seasons. The strong and continuous power spectrums shown by rainfall confirms the annual cycle variability of rainfall year-round. Sylla et al. [7] pointed out that depending on a given year, the onset of WAM may be strong or quiet in the second half of June and the West African rainfall is highly variable on intra-seasonal, interannual, and interdecadal time scales. The wavelet coherence analysis between temperature, rainfall and the four climate indices used in this study is shown below.

Wavelet coherence is a method for analysing the coherence and phase lag between two time series as both a function of time and frequency [43]. This method has been shown to be the best possible method to indicate teleconnection between two independent time series. Thus, this section focuses on investigating the teleconnection between both Conakry temperature and rainfall, and selected climate indexes. Figure 12 shows the cross-wavelet power spectra for (a) Temp–Niño, (b) Temp-AMM, (c) Temp–NA, and (d) Temp–AN, respectively. The phase relationship is represented by arrows. The regions where two cross-wavelet parameters are in phase is shown by arrows point to the right, anti-phase if the arrows point to the left, and temperature or rainfall leading (or lagging) if the arrows point upwards (or downwards), respectively. The vectors were only plotted for areas where the squared coherence is greater or equal to 0.5. More details about wavelet coherence calculations can be find in studies such as Grinsted et al. [41] and Schulte et al. [4]. The solid black line indicates the cone of influence (COI) where the edge effects become significant at different frequencies (scales), and the solid black line delimit the 95% significant regions of confidence level.

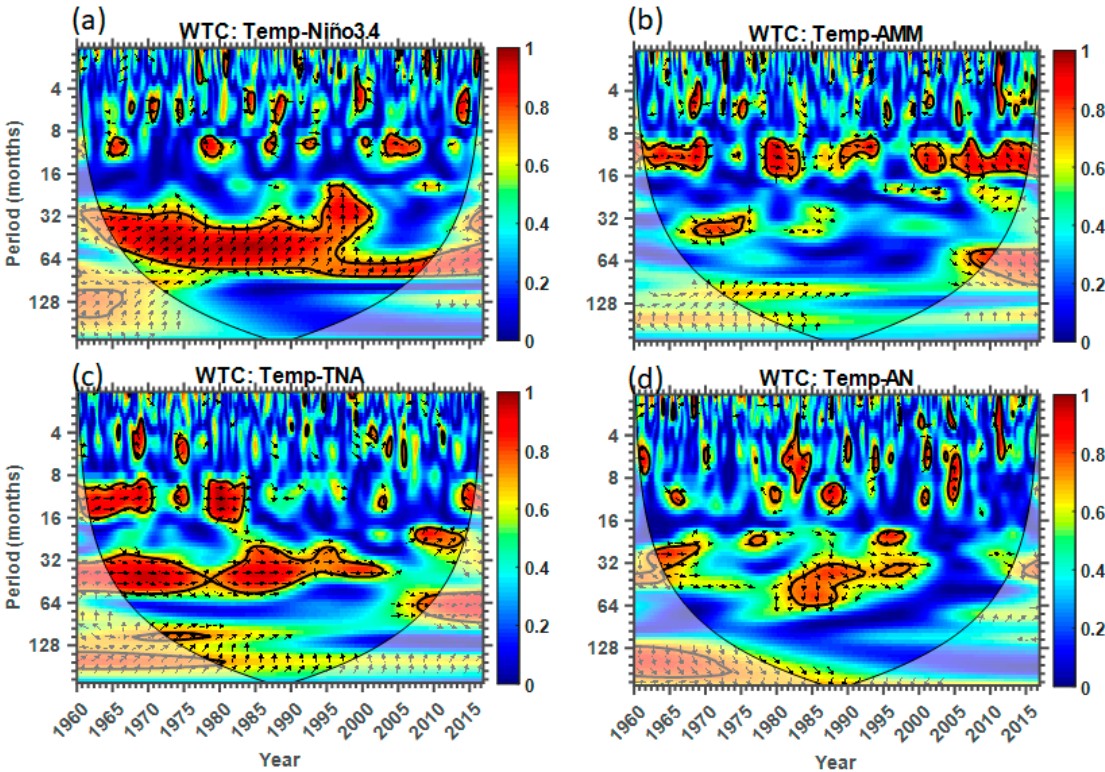

**Figure 12.** Wavelet coherence between temperature and Niño 3.4 (**a**), AMM (**b**), TNA (**c**) and AN (**d**) 1960–2016, the phase relationship is represented by arrows. The black solid line contour delimits the region (red) where the power is strong and significant and the cone of influence (solid black line) indicates the 95% confidence level.

Having found that the wavelet transform shows strong forcings with 6 and 12–months periods in temperature variability, we have proceeded to identify the wavelet coherence signature between temperature and the four climate modes (Niño3.4, AMM, TNA and AN). Figure 12a shows the

coherence calculated for the monthly mean temperature and Niño 3.4 in both time and frequency domain. At the period band of 32–64 months, significant relationship between the temperature and Niño 3.4 is clearly visible from 1960 to 2016. According to the arrows (phase) which are pointing upward and then turning to the right at the period band 32–64, the temperature seems to lead the Niño 3.4. Also, there seems to be an in-phase relationship which may indicate a strong teleconnection of the Conakry temperature to Niño variability. It is also important to note that there is a distinctive appearance of periods which are less than 13 months in the time series, with varying phase relationships between the parameters.

The wavelet coherence between temperature and AMM (Figure 12b) is observed to delineate some areas that have high significant power at periods between 4–12 months with significant peaks of distinctive periods 1960–1970, 1984–1990 and 2005–2016. A study by Foltz et al. [79] also reported that there was cooling of SSTs in the equatorial North Atlantic (ENA; 2°–12° N) in 2009 in response to a strong Atlantic meridional mode event. It is also important to mention that there are significant peaks appearing in the period band of 32–48 months during 1969–1976 and 2005–2010, respectively. A key component of the AMM is a positive feedback between the ocean and the atmosphere. Surface air pressure responds to the SST anomalies, becoming higher than normal over the anomalously cold SSTs and lower than normal over anomalously warm SSTs.

In Figure 12c, there is a significant in-phase relationship between the temperature and TNA during the period band of 8–12 months with strong power during 1962–1970, 1975 and 1973–1984. In addition, it is noted that a significant in-phase relationship is also found during 1965–2004 and 2006 at the period band of 30–64 months. The Figure 12d shows an in-phase coherence between temperature and AN corresponding at the period band of 8–12 months during 1965–1970 and 1972–1984. At the period band of 32–64 months, the AN lead the temperature, so the significant power appears at the period from 1963 to 1968, 1982 to 1992 and 1994. From these results, it can be suggested that the four climate indices contribute to drier conditions across the Conakry region. Then, the wavelet coherence spectra show that the Niño 3.4, AMM, TNA and AMM are coherent with temperature at different time scales. We can summarise that temperature is subjected to climate indices forcing at Conakry. Anomalous surface wind flow from the cold to the warm hemisphere, strengthening the mean south-easterly trade winds in the South Atlantic and weakening the north-easterly trade winds in the North Atlantic. The surface wind anomalies thus provide a positive feedback onto the initial SST anomalies by forcing changes in wind-induced evaporative cooling of the ocean.

The wavelet coherences between four climate modes and rainfall are shown in Figure 13. Significant in-phase coherence was found with the Niño 3.4 at a band period of 8–12 months (Figure 13a). This coherence suggests that the negative phase of Niño 3.4 is in agreement with dry years and its positive phase with wet years at Conakry site. The secondary peak of significant coherence appears at the band period of 16–32 months from 1962 to 1967. The observed significant coherence at a period of 16–32 months seem to be partially linked to the response of wet condition of 1960s. An in-phase relationship between rainfall and Niño 3.4 is found too at the band period of 128–256 months from 1970 to 2005. Since ENSO events can have substantial influence on African rainfall [80], the equatorial region exhibits more rainfall during El-Niño years than La-Nina years [81]. In the Indian Ocean basin, Narasimha and Bhattacharyya [82] suggested that the stronger coherence between Homogeneous Indian Monsoon and Niño 3.4 is found in the 2–7-year band and that both rainfall and the Niño 3.4 index appear irregular and random. During 1950–99, there were seven most significant El Niño events (1957–58, 1965–66, 1972–73, 1982–83, 1986–87, 1991–92, and 1997–98) for which the SST anomalies in the Niño 3 region (5° S–5° N, 150° W–90° W) exceeded 1 °C [83].

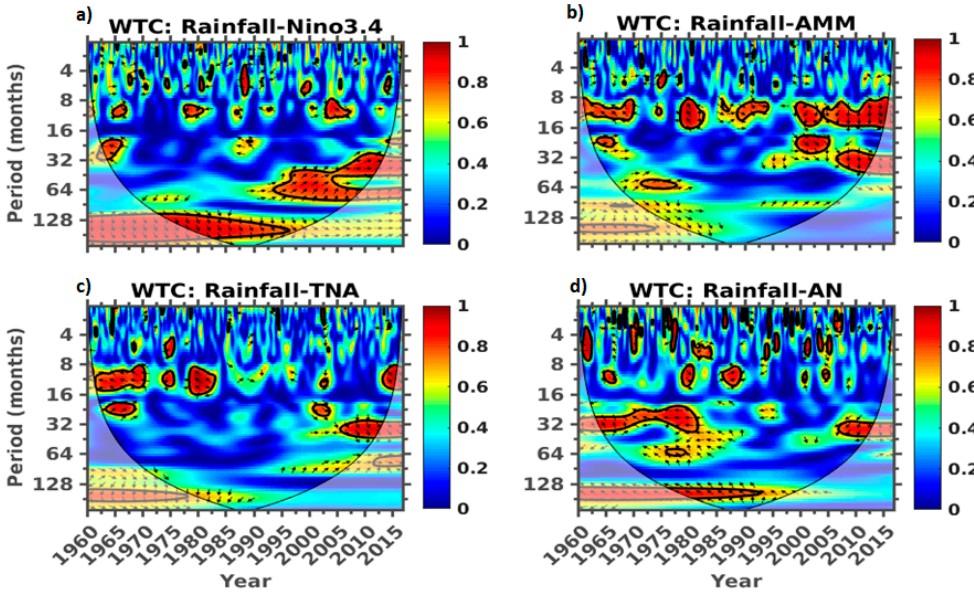

**Figure 13.** Coherence between rainfall and Niño3.4 (**a**), rainfall and AMM (**b**), rainfall and TNA (**c**) and rainfall and AN (**d**) 1960–2016, the phase relationship is represented by arrows. The black solid line contour delimits the region (red) where the power is strong and significant and the cone of influence (solid black line) indicates the 95% confidence level.

The Figure 13b depicts the wavelet coherence between rainfall and AMM and indicates that the AMM response to the rainfall variability shows an in-phase relationship at the band period of 8–12 months during the periods 1960–1970, 1978–1984, 1985–1990 and 1995–2016. The wavelet coherence analysis detected at the band period of 16–32 months lagged (i.e., AMM leading) relationship with the wet conditions during 1965, 1998–2004 and 2005–2016. Another peak is shown around 64 months from 1970 to 1984, which seems to be in relationship with the 1970s and 1980s droughts. The AMM is the dominant source of coupled ocean-atmosphere variability in the Atlantic and it affects rainfall in tropical cyclone development in the North Atlantic. During a positive phase of the AMM, the ITCZ is displaced northward. Warmer than normal SSTs and weaker than normal vertical wind shear during positive phases of the AMM tend to enhance tropical cyclone development in the Atlantic. The conditions are opposite for the negative phase of the AMM. The AMM exhibits strong variability on interannual to decadal timescales.

Figure 13c shows the coherence analysis between rainfall and TNA. A significant coherence and out-of-phase between rainfall and TNA is found at the band period of 8–12 months during 1961–1970 and 1980. A second significant coherence of in-phase relationship is shown at the band period of 16–32 months from 1965 to 1970 and 2005 to 2012. Comparing to the results of the case study for the northern part of Brazil, Uvo et al. [84] reported that the variations of April–May averaged precipitation are closely connected to the changes in the TNA SST. And sea surface temperatures in the tropical North Atlantic affect the meridional movement of the ITCZ and its band of heavy rainfall and cloud cover [73]. Furthermore, the results found by Sun et al. [85] clearly demonstrate that the climate indices have the influential consistent correlation relationship with the precipitation variation in Korea.

The wavelet coherence between AN and Conakry rainfall data was also computed (Figure 13d). The wavelet analysis detected a statistically significant coherence and in-phase relationship at the band period of 4–12 months during the years 1965, 1970, 1980, 1986 and 2005. Significant out-of-phase coherence was found at a band period of 32–48 months during the periods 1965–1984 and 2005–2013, suggesting that the positive phase of the AN contributes to drier and cooler conditions in Conakry. A period of significant coherence between the AN and rainfall extending from 1976 to 1994 was also identified at the band period of 128–190 months (~11-year), which may be due to solar cycle. Using wavelet techniques to examine the association between Indian monsoon and solar activity,

Bhattacharyya and Narasimha [86] found the power in the 8–16 y band during the period of higher solar activity at confidence levels exceeding 99.88%. The teleconnection between AN and both the temperature and rainfall measured at Conakry seems to be in agreement with previous studies (e.g., Hastenrath and Polzin. [87]; Rodriguez-Fonseca et al. [88] and others). In their study on the role of the SST anomalies in the West Africa droughts, Rodriguez-Fonseca et al., 2015 reported that the tropical Atlantic SST variability influence the West Africa rainfall in different time scales: the variability in areas closer to the equator and those at the south.

To compare our result to those of other areas, a study by Mbata et al. [89] reported for the sector of Democratic Republic of Congo that the wavelet analysis of the rainfall time series indicates an important fluctuation between practically 1960 and 1970. And then, Giannini et al. [90] suggested that the atmospheric convection and circulation changes due to the Atlantic Niño can cause increased precipitation across the equatorial Atlantic and decreases over the Sahel. These climate modes appear to have contributed substantially to the 1970s and 1980s drought in a different way and scales. The widespread influence of El Niño Southern Oscillation (ENSO) events on regional climate can have considerable socio-economic impact Climatic effects of ENSO, which vary substantially with region and season [91].

## 4. Discussion

Compared with previous studies on the variability of temperature and precipitation in West Africa [55,56], this study gives a more comprehensive investigation on variability of both warm and cool conditions that could be induced by temperature and of both wet and dry conditions by precipitation at Conakry. It provides more detailed information about their association with large-scale ocean–atmosphere oscillations as well as the trend analysis.

The findings revealed that the interannual evolution of temperature is characterized by a strong increase over the study period. As for the annual change, a semi-annual cycle and an annual cycle are found. The rainfall trend exhibits a slight decrease. Contrarily to Conakry site, Bose et al. [92] have found a significant positive increase in rainfall in the entire northern Nigeria within the period of 1970 to 2012.

Trends along the Guinea Coast are weak and non-significant except for extreme rainfall related indices, this missing significance is partly related to the hiatus in rainfall increase in the 1990s, but also to the larger interannual rainfall variability [93]. Temperature anomalies with an upward trend, remain positive since 1992 for all subsequent years, which corresponds to global continuous warming. The precipitation anomalies show a downward trend and the analysis has clearly shown the 1970s and 1980s drought periods', which caused significant material damage [94,95] and enormous loss of human life. Drought over all of West Africa is associated with the growth of positive SST anomalies in the eastern Pacific and in the Indian Ocean, and negative SST anomalies in the northern Atlantic and in the Gulf of Guinea [96]. The decrease of precipitation found by our study is in agreement with study by Aguilar et al. [97], who clearly specified that the measures of overall total precipitation are decreasing in Guinea. For the region that extend from 20° W–10° E and from 11° N–18° N, Panthou et al. [98] revealed the higher frequency of heavy rainfall and the return to wetter annual rainfall conditions since the beginning of the 2000s—succeeding the 1970–2000 drought. Furthermore, from our results, the 1970s and 1980s drought periods' have been exhibited, which were confirmed in Niger River Basin, by Djigbo F Badou et al. [55] who highlighted that the wetness of the decades, 1990s and 2000s and the manifold floods records of the first half of 2010s over West Africa are evidences that the droughts of 1970s and 1980s have stopped. An overview of the mechanisms that have been proposed to explain the influence of the AN with other climate modes within and outside the tropical Atlantic is given by Lübeckke et al. [99]. Most of part of the mechanism involve fluctuations in the wind field over the equatorial Atlantic. Part of these wind stress anomalies are excited by SST changes in the equatorial Atlantic itself [100]. They can also be due to a response to ENSO or variations in the South Atlantic subtropical high. Nnamchi et al., [101] suggested that thermodynamic feedbacks excited by

stochastic atmospheric perturbations (driving surface heat fluxes), can explain a large part of the SST variability in the eastern equatorial Atlantic. The impact of AN on rainfall over Gulf of Guinea is direct because the warm SST reduce low level wind flow inland, leading to positive precipitation anomalies over the Gulf of Guinea and adjacent coastal region [102]. After confirming the drought of 1970s and 1980s, Masih et al. [56] reported that African continent is likely to face extreme and widespread droughts in future and this evident challenge is likely to aggravate due to slow progress in drought risk management, increased population and demand for water and degradation of land and environment.

To compare our result to global temperature, from 1979 to 2016 using ERA surface air temperature, Simmons et al. [59] clarified that, early in 2016 the global temperature appears to have first touched or briefly breached a level 1.5 °C above that during the industrial area, having touched the 1.0 °C level in 1998 during a previous El Niño. Thermodynamic feedbacks constitute the main source of Atlantic Niño variability [74]. Precipitation exhibits a clear and distinct pattern during different phases of ENSO. Dynamical parameters, specific humidity and horizontal wind also exhibit clear differences for both ENSO phases [81].

The upward trend in temperature and the downward trend in rainfall was verified by the Mann-Kendall test. To understand the influence of climatic forcings on both meteorological parameters, the linear regression model was evaluated and it has been found that TNA and AMM have a more significant dependence than other indices. For extremes analysis, Aguilar et al. [97] used long term daily temperature and precipitation data set of Guinea and other countries in Africa. For Conakry station, they found inhomogeneous data before 1950 and after around 1995. Then they used RClimDex to processes them in order to get homogeneous data. But after checking the archives of database of Conakry, we found that the exceptional shift of temperature in 1998 is not linked to instrument replacement nor any error of digitalization. The wavelet analysis of both signals showed the semi-annual and annual cycles in temperature and the annual cycle in rainfall. A study conducted by Adejuwon et al. [102], for 16 stations in west Africa (Nigeria) highlighted that for all the series analyzed, there is the general tendency towards increasing aridity and spectral analysis indicates prominent periods of between 2-and 8-year cycles.

Several previous studies have shown that there is existence of significant simultaneous covariability of ENSO with West African rainfall [80,103,104]. One of the possible teleconnection mechanisms that could explain the ENSO influence on West Africa rainfall is the eastward shift of the Walker circulation and subsequent decent over the Afro-Asia during the El Niño events [104]. The process of this Walker-type circulation is associated with reduced rainfall in the West Africa during the El Niño events. Also, El Niño events also increase transport of heat flux from ocean to the atmosphere which results in tropical warming.

The observed strong relationship between AMM and both temperature (Table 1) and rainfall (Table 2) is in agreement with a study by [71]. In their study, Doi et al., [71] found a significant link between the AMM and the interannual modulation in the seasonal variation of the Guinea Dome region. This study showed that during the preconditioning phase of the positive (negative) ANN, the Guinea Dome is anomalously weak (strong) and the mixed layer is anomalously deep (shallow), there is a late fall. This means that the AMM has a strong influence in both rainfall and temperature of the Conakry, Guinea.

The variability of the SST in the tropical North Atlantic region which can produce stronger or weaker trade winds is expected to have an influent in the rainfall and temperature of Conakry. Therefore, the strong correlation that is observed between TNA and rainfall is understandable because weaker trade winds are expected to bring moisture in the Guinea coast, while stronger trade winds are expected to bring dryer conditions.

## 5. Conclusions

The semi-annual and annual cycles of temperature and the annual cycle of rainfall could be generated indeed by the ITCZ oscillation and modulated by the WAM. The models used in this study

highlighted the variability of temperature and rainfall that are characterized by a significant upward trend in temperature and a low downward trend in rainfall. The IPCC Fourth Assessment Report review of climate model projections of temperature shows a consistent warming in all subregions, but less consistent patterns for rainfall [105]. Temperature and rainfall variability at Conakry site were analysed and linked to dominant modes of climate variability at annual to multiannual timescales.

There is a strong teleconnection between the SST of both pacific and Atlantic in the variability of rainfall and temperature. So, the annual variability of temperature and rainfall at Conakry are largely influenced by climate forcings AMM, TNA and Niño3.4. However, there is no significant influence of AN on these meteorological parameters. Furthermore, the warming of 1998 seems to be a response of the 1997–1998 strong El Niño event. Physically, the influence of climate modes on temperature and rainfall were found to vary at different time scales. In Guinea, more important mining projects are under way. And, in this regard, it would also be important to take into account the anthropogenic impacts in order to deepen our knowledge. This would allow us to better understand their variability which is useful in planning sustainable development projects. It is hoped that the results from this study would help to better understand climate variability in order to get sufficient operational decision support, and resource management for a sustainable development of developing countries.

**Author Contributions:** Conceptualization, R.T.L.; Data curation, R.T.L.; Formal analysis, N.M. and N.B.; Resources, H.B.; Software, N.M.; Supervision, H.B., N.M., A.H., Z.B. and V.S.; Validation, A.H.; Writing—Original draft, R.T.L.; Writing—Review & editing, R.T.L., H.B., N.M. and V.S.

**Funding:** This research received no external funding.

**Acknowledgments:** This work is undertaken in the framework of the French South-African International Research Group LIA-ARSAIO (International Associated Laboratories—Atmospheric Research in Southern Africa and Indian Ocean) supported by the NRF and CNRS and by the Protea program. Authors are grateful to the Guinean Meteorological Service (GMS) for providing the temperature and rainfall data series used in this study, and to NOAA and KNMI web-teams for providing climate indices. Authors are thankful to Paulene Govender for proofreading. We are also thankful to the anonymous reviewers for their insightful comments.

**Conflicts of Interest:** The authors declare no conflict of interest.

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
