# Peer review of "Study on Temporal Variations of Surface Temperature and Rainfall at Conakry Airport, Guinea: 1960–2016"

_climate, doi:10.3390/cli7070093_

Round 1
Reviewer 1 Report
Reviewer Revision:
Climate 2019-474775 19 March 2019
Title: “Study on temporal variations of surface temperature 2 and rainfall at Conakry Airport, Guinea: 1960 – 2016”
Authors: René Tato Loua 1,5,6, *, Hassan Bencherif 1,4, Nkanyiso Mbatha 2, Nelson Bègue 1, Alain 4 Hauchecorne 3, Zoumana Bamba 5 and Venkataraman Sivakumar 4 5
1 Laboratoire de l’Atmosphère et des Cyclones, UMR 8105, CNRS, Université de La Réunion, Météo-France, 6 Réunion, France ; [email protected] / [email protected] 7
2 University of ZuluLand, Department of Geography, KwaDlangezwa, South Africa; 8 [email protected] 9
3 LATMOS/IPSL, UVSQ Université Paris-Saclay, Sorbonne Université, CNRS Guyancourt, France; 10 [email protected] 11
4 School of Chemistry and Physics, University of KwaZulu Natal, Durban, South Africa; 12 [email protected] 13
5 Centre de Recherche Scientifique de Conakry Rogbane, Conakry, Guinée ; [email protected] 14
6 Direction Nationale de la Météorologie de Guinée, Conakry, Guinée; 15
Abstract: The monthly averaged data time series of temperatures and rainfall without interruption of Conakry Airport (9.34°N 13.37°W, Guinea) from 1960 to 2016 were used. Inter-annual and annual changes in temperature and rainfall were investigated. Then, different models: Mann-Kendall Test, Multi-Linear-Regression analysis, Theil-Sen’s slope estimates and wavelet analysis where used for trend analysis and the dependency with these climate forcings. Results showed an increase in temperature with semi-annual and annual cycles. The findings have shown a sharp and abrupt rise in the temperature in 1998. The results of study have shown increasing trends for temperature (0.2°/decade). A decrease in rainfall (-5.7mm/decade) is found since the end of 1960s and annual cycle with a maximum value of about 1118.3 mm recorded in August in mean. The coherence between the two parameters and climate indices: El Niño 3.4 (Niño 3.4), Atlantic Meridional Mode (AMM), Tropical Northern Atlantic (TNA) and Atlantic Niño (AN), were investigated and they seem to have significant influence on temperature and rainfall at different time scales in this area. Thus, there is a clear need for increased and integrated research efforts in climate variations to reduce the negative impacts of climate change in the future.
Keywords: Temperature and rainfall anomalies; Annual and semi-annual forcing; Wavelet, inter-annual variability, Trend analysis and Climate change.
Recommendations:
1- Keywords need include the place where the study was done. Thank you,
2- Line 105: In section 2.1, it should be included the data number of average temperature, daily maximum, daily minimum and yearly temperature and the same for precipitation variable.
3- Line 152-154: The data number of all variables, shown in this interval, have to be included clearly in this section.
4- After line 158, it should be necessary indicate the software that manuscript used for the different computation tasks along the manuscript and if it is possible include some or one references. Thanks.
5- Figure 1 is composed by two panels. But it is not clear where the airport is in the first one panel. Please clarify.
6- Figure 3: Axis marks: It sees all marks with the same size. It is not correct. It should be, e.g. Fig 3c, horizontal axis the tick marks of the value indicates has be higher that the others. Also marks on OX axis should be every five units not four. It is a general rule for labels on Figures. It can see a practical example in the following publication:
“Julia Bilbao, Roberto Román, Charles Yousif, David Mateos, Argimiro De Miguel, 2014. Total ozone column, water vapour and aerosols effects on erythemal and global solar irradiance in Marsaxlokk, Malta. Atm. Env.99, 508-518.”
Please see and read the publication and observe the Figures characteristics. Please applied the method to new Figures in the manuscript. After that include the reference in the reference list of the manuscript.
7- Fig 4(a) axis OY: Mark with a tick mark the mean point between each labels with a stick smaller that the initial. Fig 4(b):axis OY, draw intervals with marks between 0 and 500 (every 100 value), the tick will be small that the 500 corresponding, and so on.
8- Fig 5a: OX axis: between each year label only include 5 years. Then include the tick marks of each year. OY axis: show the mean point between 2 temperatures with a stick,
Fig b: The same that Figure 5a.
9- Figure 6(a): Write u(t) and u´(t) near of the corresponding curve. Axis OX: include the tick marks corresponding to each year and smaller that the corresponding to multiple of 10 years. Figure 6 (b): The same that in Figure 6 (a). In Figure 6 u(t) and u´(t) are different for Temperature and precipitation, what could explain this difference?
10- Figure 7: Axis OX: In all figures between 1960 and 1965 there is 4 spaces and the correct is 5 spaces. Please correct it.
11- The Figure 8 title should include what is “n”.
12- Figure 9 (a and b): OX axis: Include the 10 tick marks of the year between 1960 and 1970 and so on. The same for the other period of years. It is important for locating maximum and minimum values.
13- Table 2: Standard error is very high, is it correct? What is the reason?
14- Figure 10: Please include the tick marks and ticks labels (as we come explain) on the horizontal OX axis in Fig 10 (a and b). OY axis: include the thick marks also in the two vertical axis.
15- Figure 11: Please include tick marks in the four horizontal axis. Please the 2 Figures should coincide, one above the other. OY axis: include 5 new marks between each one shown.
16- Figure 12: It is not clear. Also the title is not clear. Please do it new and clear.
17- It would be necessary mark the Pinatubo eruption date on the Figures, due to the Conclusions comment about its influence on 1998 warming.
18- In general, the Conclusions show little important results. The manuscript has a lot work and should obtain more specific conclusions.
19- Final Conclusions: The manuscript is interesting and the recommendations are necessary for clarify some points. After including the recommended items and prepare a new version the manuscript will be more understanding.

Reviewer 2 Report
At the indroduction a better literature review with more recent papers must be added (2016_2019).
English language must be improved for better undrestanding and be easier to read.
Also, a more detail discussion and comparison with results from other similar research is necessary.
Author Response
The response to your comments ares in the word draft

Reviewer 3 Report
Even if the topic is of great interest, I did not review the overall manuscript, I stopped at page 12 out of 27. I cannot review correctly the current version. The current version must be deeply revised and shortened.
The keywords are not suitable.
The manuscript is too long. It can be easily shortened e.g. the sections 2.2.1. The MK test is widely used and a full description of statistics is not relevant. In addition, a lot of sections are not useful e.g. L414-419. The current version needs to be shortened by at least 40%.
The manuscript must be deeply reorganized i.e. plenty of information in section “Results” should be moved to “Discussion” e.g. L289-293, L361-365 or “Materials and Methods” e.g. L420-424, L469-471, L558-560.
I suggest analyzing the Tmax trends over time. The Tmax parameter is important e.g. the number of days exceeding 30°C is expected to occur more frequently.
Author Response
Please let find the response to your comments in the Word file.

Reviewer 4 Report
General comments:
In this study, temporal variations of surface temperature and rainfall at Conakry Airport are investigated by using monthly observation data. At the same time, relationships between the above two meteorological variables and several climate indices are also examined. However, the authors just showed results of fundamental characteristics of temporal variations (seasonal variations, annual cycle, inter-annual variations), trend analyses, and wavelet analysis. The authors referred many past studies, but there were no specific discussions about the results shown in this manuscript, and the reviewer could not find any new findings or ideas about climate or meteorological systems around the western Africa. The authors are expected to make deeper consideration about results. At the same time, structure of the manuscripts and presentation of results are poor.
Thus, the reviewer cannot support publication of the manuscript and decide to reject it. I hope the following comments to be help of your work.
Major comments:
l The authors showed results by multiple analysis methods (Mann-Kendall’s trend test, MLR, wavelet analysis), but the descriptions were quite superficial. Fundamental seasonal variations and inter-annual variations of meteorological variables (e.g. temperature, rainfall, etc.) can be found in a textbook. Deeper considerations of results are indispensable to reveal climate system and mechanisms around the west Africa.
Although the authors introduced many related works just after the analysis results, it is difficult to find relationship or analogy between the past works and results shown in the manuscript. Appropriate explanations of mechanisms or climate systems should be added based on the results in this work.
l There are some questions in method (or application of each analysis method). Details will be shown below in minor comments.
l As stated above, presentations are poor. The authors have to be careful to prepare manuscript following the instruction of the journal and fundamental manners to write research papers.
l English proofreading by a professional service is strongly recommended before submission of the manuscript.
Minor comments:
- L.18-19: “The monthly … were used.” for what? The purpose is not clear.
- L.30-31: “negative impacts of climate change” cannot find before this sentence. This conclusion is too sudden.
- L.36: Climate system is very complicate. The reviewer cannot agree with the description (average and extremes define climate).
- L.55-57: Why is the target period different (1906-1992 and 1949-1990)?
- L.169: Are the overall mean values (26.5 degC and 3806.8mm) monthly? or Annual? Description is not enough.
- L.272-293: Effects of ENSO and the Mt. Pinatubo were referred here, but there is no specific discussion about the mechanism how ENSO or Mt. Pinatubo make warmer condition around Conakry. Deeper consideration is indispensable.
- L. 294: What is “regime”? Clearly identify the “regime”.
- L.329-347: Very fundamental meteorological process is described here. Nothing new, and we can find them in a textbook. Such explanation is not necessary.
- L.397: The author cannot understand the value “-2.5mm”. Is this anomaly from annual mean rainfall? If so, -2.5mm is too small. Fig.3d shows the range of variations in the annual rainfall is several hundred to 1,000 mm. Confirm the unit or give detail explanation of results in Fig.5d.
- L.414: Was the Mann-Kendall test applied to monthly data? It is better to reconsider of application of trend test using monthly data. If long-term trend is examined by using monthly data, effects of seasonal cycle could be included in the result. Usually, long-term trend is investigated by using annual mean values or annual variations of the target month (air temperature in July, etc.).
- L.441: What is “the trend in 1961”? Trend could be calculated from temporal variations in multiple years. Or, does this mean “upward trend from January to December in 1961”? The authors carefully review the definition of trend and appropriateness of the application in this study.
- L.460: The reviewer cannot agree that drought forecasting will be improved by SQ-MK model. It just examines the trend in the past, but cannot predict future trend. If it can, please give additional description.
- Fig.8: Why are there many correlations between two indices (many dots in the lower left panels)?
- L.527-532, Fig.9: What is the point of Fig.9? Just showing the results is not enough. (Same for L.551-553, Fig.10)
- L.559: The Theil-Sen method is applied with the monthly data. Is it OK? As stated above about Mann-Kendall’s test, annual mean or values in a specific period in each year may be better to use. Please reconsider the application.
- L.587: The reviewer cannot agree “increase in heavy rainfall”. Increasing total rainfall never promise increasing heavy rainfall.
- L.597-606: Peaks around 6 and 12 months indicate simple seasonal cycle, but not special phenomena. The authors have to be more careful to interpret results.
- Results of wavelet analysis (P.19-21): The authors just described the period of significant peak. Why are these peaks found? What is the mechanism between the variations in temperature (or rainfall) and each index? Not only superficial description but deep discussion is necessary.
- L.731: What is “Findings”? And a semi-annual and annual cycle is very fundamental of meteorological process, so these are not revealed in this study. Overall, detail discussion is missing in this study.
- L.740: Frequent and long drought could not find in the result of this study.
- L.769-773: These sentences are too sudden in this context.
- L.782-783: The authors tried to find any relationships between meteorological conditions and several climate indices, but nothing is clear or no possible process or mechanisms are proposed in this study. Detail investigations for each results are necessary.
- L.793-797: The results in this work are too far to improve meteorological forecast of climate projections. The authors should reconsider the position of this work in the context of climate researches.
Author Response
The responses are in the word file.

Round 2
Reviewer 3 Report
Abstract - Remove all acronyms (L27-28). Keywords - "Temperature and rainfall anomalies" & "Annual and semi-annual forcing" are not suitable, please change it. Results - the titles of sections 3.2.1, 3.2.2 & 3.2.3 are not suitable for a section "results". I suggest joining the 3 sub-section under only one section 3.2 about "trends analysis". The section 3.2 should be strongly shortened. For instance, the section 3.2.1 can be summarized as 3 lines "we observed an increase/decrease of rainfall by ... over the time-period...".
Author Response
The responses of reviewer 3 comments are in the attached file.

Reviewer 4 Report
Attached please fing the review comments.

Author Response
Please the response to the comments are in the attached file.
